# Total Neoadjuvant Treatment for Locally Advanced Rectal Cancer Patients: Where Do We Stand?

**DOI:** 10.3390/ijms241512159

**Published:** 2023-07-29

**Authors:** Valentina Daprà, Marco Airoldi, Michela Bartolini, Roberta Fazio, Giuseppe Mondello, Maria Chiara Tronconi, Maria Giuseppina Prete, Giuseppe D’Agostino, Caterina Foppa, Antonino Spinelli, Alberto Puccini, Armando Santoro

**Affiliations:** 1Department of Biomedical Sciences, Humanitas University, 20072 Pieve Emanuele, Italy; 2Medical Oncology and Hematology Unit, Humanitas Cancer Center, IRCCS Humanitas Research Hospital, 20089 Rozzano, Italy; 3Radiotherapy and Radiosurgery Department, IRCCS Humanitas Research Hospital, 20089 Rozzano, Italy; 4Division of Colon and Rectal Surgery, IRCCS Humanitas Research Hospital, 20089 Rozzano, Italy

**Keywords:** locally advanced rectal cancer, total neoadjuvant therapy, rectal cancer, nonoperative management, ct-DNA, MMR-deficient tumor

## Abstract

The therapeutic landscape in locally advanced rectal cancer (LARC) has undergone a significant paradigm shift in recent years with the rising adoption of total neoadjuvant treatment (TNT). This comprehensive approach entails administering chemotherapy and radiation therapy before surgery, followed by optional adjuvant chemotherapy. To establish and deliver the optimal tailored treatment regimen to the patient, it is crucial to foster collaboration among a multidisciplinary team comprising healthcare professionals from various specialties, including medical oncology, radiation oncology, surgical oncology, radiology, and pathology. This review aims to provide insights into the current state of TNT for LARC and new emerging strategies to identify potential directions for future research and clinical practice, such as circulating tumor-DNA, immunotherapy in mismatch-repair-deficient tumors, and nonoperative management.

## 1. Introduction

Colorectal cancer (CRC) is the third most frequently diagnosed cancer worldwide. Rectal cancer is estimated to represent 31% of CRC cases. Between 5% and 10% of patients with rectal cancer present with a locally advanced disease at diagnosis, which is usually managed through multimodal therapy, including radiation, chemotherapy, and surgery [1,2,3]. For many years, long-course chemoradiotherapy (CRT) or short-course radiotherapy (SCRT), both followed by total mesorectal excision (TME) surgery, have been deemed as standard neoadjuvant treatments for stage II–III rectal cancer [4,5]. LCRT consists of 45–50 Gy in 25–28 fractions with concurrent chemotherapy, either fluorouracil or capecitabine. Based on evidence from the German trial by Sauer et al., this regimen is considered superior to postoperative chemoradiotherapy in terms of local control, with a 5-year cumulative incidence of local recurrences of 6% in the preoperative and 13% in the postoperative arm [6]. On the other hand, SCRT involves the delivery of five fractions of 5 Gy followed by surgery, either within 1 or 8 weeks, and the value of this combination of treatments in reducing the risk of local recurrence has been corroborated by several studies [7,8,9,10,11]. Two randomized studies showed that when compared, LCRT and SCRT are equivalent in terms of survival, local control, or late toxicity. However, SCRT was associated with a clear reduction in complete pathological responses (pCR) and an increase in the percentage of circumferential margin infiltration (CRM) and local recurrences [4,5]. Therefore, despite the advantages in terms of economic health and quality of life associated with SCRT, the benefit/harm ratio was ultimately beneficial to LCRT. Specifically, both clinical practice and international guidelines recommend LCRT when substantial tumor downsizing is needed, either to increase R0 resection rate or to allow sphincter-sparing surgery [12,13].

On the other hand, the role of adjuvant chemotherapy after preoperative LCRT/SCRT remains controversial and has not been shown to improve overall survival (OS), partly because of poor compliance to treatment [14,15,16]. Hence, international guidelines advocate for adjuvant chemotherapy solely in patients with yp stage III (and ‘high-risk’ yp stage II) following preoperative LCRT/RT [12]. Nevertheless, a substantial proportion of patients (25% to 30% at 5 years) suffer tumor distant recurrence [17]. Given the debatable role of adjuvant chemotherapy, administering all the systemic treatment before surgery appeared to be an appealing approach.

The introduction of TNT, consisting in upfront chemotherapy, followed or preceded by either LCRT or SCRT, has presented a recent paradigm shift in the management of locally advanced rectal cancer (LARC). Over the last decade, the lack of large randomized studies regarding the use of TNT has generated substantial heterogeneity in terms of recommendations from international guidelines and treatment models from various centers. Here, we present a timeline (Figure 1) and a selection of the most significant phase III trials in this setting.

## 2. Phase III Practice-Changing Trials

### 2.1. RAPIDO

In the RAPIDO trial, 912 patients were randomized to receive a preoperative treatment consisting in SCRT followed by chemotherapy (six cycles of CAPOX or nine cycles of FOLFOX) and TME, compared with standard-of-care treatment (Figure A1). All the patients enrolled had high-risk stage II and III rectal cancer, defined as having at least one of the following criteria at MRI staging: cT4a, cT4b, extramural vascular invasion (EMVI), cN2 (metastasis in four or more locoregional lymph nodes), involved mesorectal fascia (MRF), or enlarged lateral lymph nodes [18] (Table 1). 

The primary endpoint was 3-year disease-related treatment failure (DrTF), which included locoregional failure, distant metastasis, new primary colon tumor, or treatment-related death. Patients receiving TNT demonstrated a decreased DrTF rate of 23.7% compared with 30.4% in the control arm (HR 0.75, *p* = 0.019). The experimental arm also showed doubled pCR rate (28.4% vs. 14.3%) and significantly lower rate of distant metastases (20.0% vs. 26.8%, *p* = 0.005). However, in the neoadjuvant phase, the use of TNT was associated with higher rates of grade ≥ 3 adverse events (48% vs. 25%, respectively), leading to treatment discontinuation rates of 15% vs. 9%, respectively. Given the longer systemic treatment delivery in the standard-of-care arm, the global rates of severe adverse events were 63%, compared with 48% in the control arm [18].

A recent update of RAPIDO, with a median follow-up of 5.6 years, showed that the experimental treatment was associated with an increased risk of locoregional recurrence (LRR) compared with the standard of care (12% vs. 8% respectively, *p* = 0.007), while the reduction in DrTF and distant metastases was sustained. The 5-year OS rates were 81.7% vs. 80.2%, respectively (Table 2) [19].

### 2.2. PRODIGE-23

The PRODIGE-23 trial randomized 461 patients with rectal adenocarcinoma and high-risk features, such as cT3/cT4 disease (Table 1), to receive six cycles of mFOLFIRINOX, followed by LCRT, TME, and a further six cycles of adjuvant mFOLFOX versus long-course standard LCRT, followed by TME and twelve cycles of mFOLFOX6 or eight cycles of capecitabine (Figure A1) [20].

The primary endpoint of the PRODIGE-23 trial was disease-free survival (DFS), whereas toxicity, pCR, metastasis-free survival (MFS), OS, and quality of life were secondary endpoints. Patients receiving TNT exhibited an improved 3-year DFS of 76% compared with 69% in the standard-of-care group (*p* = 0.034). The experimental arm also showed a doubled pCR rate (27.8% vs. 12.1%) and increased 3-year MFS (79% vs. 72%) [20]. In total, 98% of the patients in the TNT group received mFOLFIRINOX, and 92% of them managed to complete the preplanned six cycles. The overall incidence of grade 3 or higher adverse events in these patients was 46%, which is similar to the control arm. Of note, the reduced duration of adjuvant chemotherapy in the TNT group led to less severe toxicity, including peripheral sensory neuropathy. The regimen appeared to be better tolerated than in previous studies that analyzed FOLFIRINOX in metastatic setting, due to the omission of fluorouracil bolus and the inferior dose of fluorouracil in continuous infusion [21,22].

The 7-year results of PRODIGE-23 were presented during the 2023 ASCO conference and indicated that the TNT arm achieved a 7-year DFS rate of 67.6%, compared with 62.5% in the standard-of-care arm. Increases in MFS, cancer-specific survival, and OS were also observed. Among patients with metastatic relapse, the median OS was 44.4 months in the neoadjuvant arm, compared with 39.4 months in the standard-of-care arm (*p* = 0.062) (Table 2) [23].

### 2.3. PROSPECT

Data from the randomized phase III noninferiority PROSPECT trial were recently presented during the 2023 ASCO Annual Meeting Plenary Session. A total of 1128 patients with staged cT2N+, cT3N0, or cT3N + LARC were randomly assigned to receive either six cycles of neoadjuvant mFOLFOX6 and selective LCRT (experimental arm) or standard preoperative LCRT (control arm) (Table 1). Both treatment arms involved TME and adjuvant chemotherapy (Figure A2). The primary endpoint of the trial was disease-free survival (DFS), while secondary endpoints included time to local recurrence, OS, complete (R0) resection rate, complete response rate, and toxicity. 

A total of 585 participants were enrolled in the experimental group and underwent radiological restaging after six cycles of mFOLFOX6. If the tumor shrank by 20% or more based on radiological assessment, no radiation therapy was given before surgery. Participants with suboptimal responses (<20% tumor regression) or who were unable to tolerate at least five cycles of chemotherapy received LCRT before surgery. Among the experimental group, 53 out of 585 patients (9%) required LCRT before surgery; of those, 38 out of 53 patients (6.5%) did not reach the clinical response threshold of 20% reduction.

After a median follow-up of 58 months, the 5-year DFS rate for the experimental arm was 80.8%, compared with 78.6% in the control arm (HR 0.92, CI [0.74, 1.14]; stratified noninferiority *p* = 0.0051), meeting the criterion for noninferiority. The 5-year OS was 89.5% in the experimental arm, compared with 90.2% in the control arm. Local recurrence rates were very low and similar for both groups (2%). pCR rate was 22% in the experimental arm and 24% in the control arm (Table 2). During the neoadjuvant period, more participants in the FOLFOX group experienced neuropathy, while more participants in the control group had diarrhea. Preoperative ≥ grade 3 adverse events occurred more frequently in the experimental arm (41.0% vs. 22.8% in the control arm), whereas the control arm had a higher occurrence of postoperative ≥ grade 3 adverse events (39.0% vs. 25.6% in the experimental arm) [24]. At 12 months after surgery, patients in the FOLFOX group reported significantly lower rates of fatigue and neuropathy, as well as improved sexual function compared with the control group (all multiplicity adjusted *p* < 0.05). However, there were no significant differences between the two groups in terms of bladder function or health-related quality of life at any time point [25].

### 2.4. Considerations

Since 2020, the compelling outcomes from the RAPIDO and PRODIGE-23 trials have had a significant impact on clinical practice, establishing a new standard of care. The use of a TNT approach is now considered the preferred option for stage II-III LARC, particularly when high-risk features are present. However, data from the PROSPECT trial have demonstrated that neoadjuvant chemotherapy alone can be effective in treating low/intermediate risk LARC, allowing for the omission of subsequent chemoradiotherapy (CRT). As a result, for low-risk tumors, a less intensive treatment approach may be considered acceptable.

PRODIGE-23 was the only trial that included adjuvant chemotherapy regardless of ypTNM stage, splitting systemic treatment in a perioperative approach. One reason for this approach could be to maximize patients’ tolerance to mFOLFIRINOX treatment and to lower the risk of perioperative complications. Only 1 patient out of 226 discontinued neoadjuvant treatment because of drug-related toxicity.

It is noteworthy that despite the impressive rates of pCR, only the PRODIGE-23 trial was able to demonstrate an OS benefit in the TNT arm. Indeed, pCR is known to be a poor surrogate endpoint for OS at trial-level surrogacy in rectal cancer [26]. On the other hand, DFS, which was the primary endpoint of the PRODIGE-23 trial, is a stronger predictor of OS than pCR, as shown in a meta-analysis [27]. Median OS data from other phase III trials are eagerly awaited.

## 3. How to Decide Optimal Treatment?

### 3.1. Patient Selection Criteria

Defining the patient population that stands to derive the greatest benefit from this therapeutic approach is crucial. The TNT approach has long been regarded as a viable option, particularly in locally advanced stages of the disease where there exists a higher risk of metastatic spread and distant recurrence [12]. The presence of high-risk features was the main eligibility criterion for pivotal trials of TNT [28,29,30]. For these reasons, patients recruited in the RAPIDO trial must have had one of the following five high-risk features at baseline pelvic MRI: T4, N2, EMVI+, MRF+, or enlarged lateral pelvic lymph nodes. A high proportion of patients in this study (up to 65%) had at least two risk factors, accounting for a poor prognosis [18]. However, in the PRODIGE-23, all stage II–III patients (as revealed by MRI +/− endorectal ultrasound) were recruited regardless of the presence of high-risk features. A total of 90% of patients had cN+ tumors at baseline, but compared with RAPIDO, there was a lower proportion of those with T4 stage (17% vs. 31%) and MRF involvement/threatening (27% vs. 61%) [20]. 

Differently from the above-mentioned studies, the PROSPECT trial focused on low–intermediate-risk patients, staged as cT2N1, cT3N0, or cT3N+ mid/high LARC. Exclusion criteria were cT4 staged tumor, a low-lying distal tumor that would have required an abdominoperineal resection (APR) at baseline, or more than four enlarged nodes (cN2) [24].

Treatment selection should not solely rely on tumor-related factors but also take into account patient-specific characteristics and age. The vast majority of patients (~80%) enrolled in the RAPIDO and PRODIGE-23 trials had an ECOG performance status of 0. The median age in both trials was 61 years; only 12% of patients in the PRODIGE-23 trial were ≥70 years old (Table 1). Therefore, it appears that TNT has been validated for younger and fit patients. However, it is highly likely that the findings from the RAPIDO and PRODIGE-23 trials may not be applicable to an older and less physically fit real-world population. 

### 3.2. Alternative Schemes of TNT in Phase II Trials

#### 3.2.1. GRC-3

The GRC-3 phase II trial randomized 108 patients to receive either conventional LCRT followed by TME and four cycles of adjuvant CAPOX regardless of response or induction chemotherapy (four cycles of CAPOX) before LCRT and TME. The primary results were reported in 2010 [31]. The arm receiving induction chemotherapy before LCRT demonstrated similar rates of pCR (~14%) and complete resection (R0) (~86%), while achieving more favorable compliance and toxicity profiles. Five-year DFS rates were comparable between the two treatment arms (64% and 62%), as were OS rates (78% vs. 75%). However, the study design was not powered to detect subtle differences in these outcomes. Among the patients who underwent surgery, the achievement of T downstaging and adherence to the prescribed adjuvant/induction chemotherapy regimen (i.e., receiving three to four cycles out of the total planned cycles) demonstrated a significant correlation with a reduced risk of disease relapse. After a median follow-up of 69 months, the data support the feasibility of the induction strategy, since it did not compromise DFS, OS, or the cumulative incidence of local and distant relapse and it was associated with better treatment compliance and lower acute toxicity [32].

#### 3.2.2. CAO/ARO/AIO-12

The CAO/ARO/AIO-12 study enrolled 306 patients with stage II–III rectal cancer who were randomized to receive oxaliplatin-based chemotherapy (3 cycles of FOLFOX) either before (induction chemotherapy arm) or after (consolidation chemotherapy arm) oxaliplatin-based LCRT [33]. TME was scheduled 18 weeks after the initiation of TNT in both groups, regardless of tumor response. The primary endpoint of the study was the rate of pCR. The authors used a pick-the-winner design on the basis of the hypothesis of an increased pCR of 25%. The results show that patients in the consolidation chemotherapy arm had a higher pCR rate compared with those in the induction chemotherapy arm (25% vs. 17%, respectively); therefore, only the consolidation arm met the predefined statistical hypothesis [33,34]. Furthermore, the consolidation chemotherapy arm demonstrated better compliance with LCRT but inferior compliance with chemotherapy. The 3-year DFS for both groups was 73%. The 3-year incidences of distant metastasis, locoregional recurrence, and OSrates were similar in both treatment groups. Results of a secondary analysis did not reveal a significant difference in long-term oncologic outcomes, chronic toxicity, patient-reported outcome measures for global health status and quality of life, or stool incontinence between the two groups [35].

#### 3.2.3. OPRA

The OPRA trial was a phase II randomized trial that enrolled 324 patients with LARC, requiring abdominoperineal resection or coloanal anastomosis at baseline. Patients were randomized to receive LCRT with either induction (INCT) or consolidation chemotherapy (CNCT), consisting in eight cycles of FOLFOX or six cycles of CAPOX. Tumor restaging was performed within 8 (±4) weeks after TNT: patients with incomplete clinical response were recommended TME, while patients with complete clinical response (cCR) or near cCR were offered participation in a standardized watch-and-wait (WW) protocol [36].

Grade 3 to 4 toxicity occurred similarly between the two treatment groups. 

Of the 304 patients restaged after TNT, 79 (26%) were recommended to undergo TME: 41 of 146 (28%) in the INCT-LCRT group and 38 of 158 (24%) in the LCRT-CNCT group.

Among the 225 patients offered WW, 81 (36%) developed a regrowth during the follow-up and subsequently underwent salvage TME [37]. Of note, almost 10% of patients undergoing salvage TME had a pCR on final specimen examination [38]. 

The 3-year DFS was 78% for the CNCT and 77% for the INCT arm. The 5-year DFS was 71% and 69%, respectively; the 5-year OS was 88% in both arms [37]. The proportion of patients with preserved rectum at 3 years was 60% for the CNCT and 47% for the INCT arm. At 5 years of follow-up, these rates were 54% and 39%, respectively. The rates of sphincter-preservation did not differ between the two treatment groups, and DFS after TME was comparable regardless of whether TME was performed after restaging or at the time of regrowth. 

Although not formally comparable, these findings indicate that LCRT followed by CNCT was associated with a higher rate of organ preservation, thereby leading to higher 3-year and 5-year TME-free survival [36,37].

### 3.3. Induction vs. Consolidation Chemotherapy

Upfront LCRT followed by CNCT resulted in sustained cCR/organ preservation (OPRA) or higher pCR (CAO/ARO/AIO-12), without affecting DFS and metastases-free survival [33,36,37]. The main contributing factor to the observed effect of consolidation chemotherapy is most likely the longer interval between completion of LCRT and restaging or surgery. This extended timeframe could allow for a more profound impact of radiation therapy, without increasing surgical morbidity.

In the CAO/ARO/AIO-12 trial, the median interval from the end of CRT to surgery was 45 days in the induction arm and 90 days in the consolidation arm [33]. Similarly, in the OPRA trial, the median interval from the end of CRT to restaging was 8.0 weeks in the induction arm and 28.5 weeks in the consolidation arm, due to the different duration of the chemotherapy protocol employed [36].

Although the order of chemoradiation and systemic chemotherapy does not impact disease-free survival, the optimal sequence for TNT remains a topic of ongoing discussion, as it may be influenced by different pretreatment tumor characteristics or treatment objectives. INCT and the associated superior chemotherapy compliance could be effective in addressing micrometastatic disease, such as EMVI-positive or lymph-node-positive tumors, or those associated with high serum carcinoembryonic antigen (CEA) levels.

On the other hand, LCRT followed by CNCT may enhance local tumor regression, enabling curative resection in challenging scenarios like cT4 tumors or those involving the mesorectal fascia. It may also help avoid abdominoperineal resection in low-lying tumors and to enhance the probability of organ preservation. In the OPRA trial, more than half of the patients that received LCRT-CNCT achieved this outcome and maintained the benefit, being still TME-free at the 5-year follow-up [37].

The PROSPECT trial findings showed an additional and intriguing perspective on the potential curative intent of induction chemotherapy alone for low/intermediate-risk LARC. This approach could offer advantages by sparing patients from the long-term toxicities associated with subsequent RT. It may be particularly beneficial for younger patients seeking fertility preservation or desiring to avoid early menopause [24].

### 3.4. Risk of Overtreament

The purpose of proper preoperative staging is to facilitate treatment planning by identifying patients who are more likely to benefit from preoperative therapy and to aid in determining the optimal surgical procedure. Consequently, it is crucial to minimize the adverse consequences, side effects, and costs of unnecessary preoperative treatment when a patient lacks adverse prognostic features. A retrospective study from Lord et al. examined 378 patients in the United Kingdom who underwent upfront surgery for rectal cancer without preoperative radiotherapy [39]. Since 2020, the new National Institute for Health and Care Excellence (NICE) guidelines, in alignment with the NCCN guidelines, have recommended preoperative radiotherapy for all patients, except those with radiologically staged T1–T2, N0 tumors [13,40]. The study aimed to assess the impact of these guidelines by evaluating the oncological outcomes of the treated patients and stratifying them based on MRI high-risk criteria. The findings revealed that if the 2020 NICE guidelines were implemented, 248 patients (66%) who underwent primary surgery would have received preoperative radiotherapy, despite demonstrating a promising local recurrence rate of 6% with surgery alone. However, if MRI high-risk criteria were utilized instead of NICE criteria, only 121 patients (32%) would have undergone preoperative radiotherapy.

Another prospective study conducted in Germany, involving a cohort of 878 patients, demonstrated that the combination of high-quality MRI and TME surgery, followed by a standardized examination of the resected specimen, could potentially allow for the omission of preoperative LCRT in over 40% of patients with stage II or III rectal cancer. This approach was found to carry a minimal risk of undertreatment [41].

### 3.5. The Role of Surgery

Surgical management depends on patient characteristics, tumors factors, and response to neoadjuvant therapies. The aim is to optimize function and survival with the lowest risk of recurrence. The clinical assessment of rectal cancer by a colorectal surgeon is essential before and after treatment in order to determine the management in a multidisciplinary context. In fact, through digital rectal examination, the surgeon can add important information to the endoscopic and radiologic evaluation: the location of the tumor (anterior, posterior, or lateral), mobility, proximity to the sphincter, and condition of the sphincters. Surgical options after neoadjuvant treatment may range from a rectal resection with TME to organ-sparing techniques, such as local excision of the residual tumor (in very selected cases). The primary goal of rectal cancer surgery is to achieve a complete excision of the tumor and surrounding mesorectum with tumor-free margins (R0 resection). This endpoint has been proven to have the greatest effect on recurrence and OS [42]. TME, in which the primary tumor is resected along the embryological fascial planes with all the associated lymphatics, is the gold standard for curative resection, decreasing local recurrence rates and improving survival [43]. In patients in whom a restorative procedure is not possible owing to tumor location, invasion, or involvement of the sphincter complex that may impair continence, an abdominoperineal resection might be the best surgical option. In the absence of these features, sphincter preservation with a restorative proctectomy is usually feasible. In restorative proctectomy after a neoadjuvant treatment a defunctioning stoma is usually performed to reduce the morbidity associated with an anastomotic leak (AL). AL is a potentially life-threatening complication, significantly impacting mortality, short- and long-term morbidity, quality of life, and local recurrence rates. Although its etiology is multifactorial, there are some risk factors related to surgery such as a double-stapled anastomosis and a suboptimal colonic vascularization. In this context, a single-stapled anastomosis seems to reduce the AL rates [44]. Colonic blood supply assessment with indocyanine green real-time angiography seems to reduce AL rates in rectal cancer surgery [45], although results from the European prospective multicenter IntAct RCT are eagerly awaited [46].

## 4. Discussion and Future Perspectives

### 4.1. Role of Microsatellite Instability (MSI)

Among all patients diagnosed with rectal cancer, microsatellite instability (MSI) or mismatch repair-deficient (dMMR) tumors can be found in approximately 3% of the patients [12]. MSI is a consequence of a deficient mismatch repair system that results in the accumulation of insertion and/or deletion mutations within microsatellite DNA regions. dMMR can either result from the inheritance of a germline mutation in an MMR gene (e.g., *MLH1*, *MSH2*, *MSH6* or *PMS2*) responsible for Lynch Syndrome (LS) or, more commonly, from the epigenetic inactivation of *MLH1* in sporadic cases [47,48,49]. Patients with MSI-H/dMMR locally advanced CRC have unique clinicopathological and molecular characteristics and typically experience slow clinical progression. Furthermore, specific characteristics like higher tumor-infiltrating lymphocytes and overexpression of immune checkpoint receptors and ligands, mainly PD-1 and PD-L1, give them special sensitivity to immunotherapy, as suggested by the emerging evidence outlined below [50].

A seminal phase II study conducted by Cercek et al. supports the predictive role of MSI for better responses to immunotherapy. In this study, 36 patients with locally advanced rectal adenocarcinoma and microsatellite instability were treated in the neoadjuvant setting with the anti PD-1 monoclonal antibody dostarlimab. The analysis focuses on two primary endpoints: first, sustained clinical complete response or pCR after completion of dostarlimab therapy with or without chemoradiotherapy; second, overall response to neoadjuvant dostarlimab therapy with or without chemoradiotherapy. A total of 23 patients completed treatment with dostarlimab and underwent a follow-up of at least 6 months. All 23 patients had a clinical complete response, with no evidence of tumor. None of the patients underwent surgery or received chemoradiotherapy, and no cases of progression or recurrence were reported during follow-up (range 0–36.3 months) [51].

Another retrospective analysis conducted by Cercek et al. identified fifty patients with dMMR rectal cancer treated in the neoadjuvant setting and compared them to a matched proficient MMR (pMMR) rectal cancer cohort [52]. Among these patients, 21 (42%) were treated with TNT approach consisting in induction FOLFOX as initial treatment followed by LCRT, 16 (32%) received neoadjuvant LCRT, and 13 (26%) proceeded directly to surgery. The results show that the 21 dMMR patients treated with total neoadjuvant had an early progression rate of 29% (6 of 21 patients) during induction with FOLFOX. In comparison, no progression was noted in 63 pMMR rectal tumors who were treated with the same regimen (*p* = 0.0001). The 16 patients with dMMR tumors treated with neoadjuvant chemoradiation achieved tumor downstaging in 93% of cases, with 14% pCR (similar to the 17% obtained in the historical cohort of 48 patients with pMMR tumors). These results support the notion that dMMR patients have a poorer response to neoadjuvant chemotherapy than pMMR patients. Notably, some patients with progression of disease on neoadjuvant chemotherapy achieved a subsequent response to chemoradiation, suggesting normal sensitivity to LCRT [52]. Therefore, the role of MSI-H/dMMR in rectal cancer can be described either as a negative predictor of efficacy of neoadjuvant chemotherapy or as a positive predictor of preoperative immunotherapy effectiveness. Although data come from a single study with a limited sample size, they still provide strong evidence of a positive short-term efficacy of preoperative immunotherapy as a single treatment modality in patients with LARC and MSI. These promising results provide a framework for the evaluation of highly active anticancer therapies in the neoadjuvant setting, where patients would avoid chemoradiotherapy and surgery as their tumor is treated when it is most likely to respond, namely before exposure to other agents that could allow rapid progression of the tumor or select for cells with a resistant phenotype. Moreover, as many LARC patients have bulky and clinically symptomatic tumors, prompting local control with selection of the optimal first-line treatment is imperative. Therefore, considering the strong implications on the choice of therapeutic strategy, the evaluation of the status of mismatch repair proteins in patients with locally advanced rectal carcinoma should be requested and available before the start of a neoadjuvant treatment.

Finally, finding dMMR in CRC triggers the detection of a potential heritable germline deficiency in MMR. Identification of dMMR is more frequently linked to LS in rectal cancer than in colon cancer, with an incidence of LS of 84% in MSI rectal cancer [51]. Patients with identifiable pathogenic mutations have LS, which is important to identify not only for specific implications regarding the treatment approach as previously mentioned but also for the enrolment in specific screening programs for CRC, as well as gynecological cancers, cascade testing, and prevention for family members.

### 4.2. Nonoperative Management

The introduction of TNT has increased the rates of both clinical and pathological complete response, resulting in excellent long-term oncological outcomes. As a result, nonoperative management (NOM) of patients with a cCR after neoadjuvant therapy has gained acceptance as a potential treatment option in selected cases [38]. NOM is based on the replacement of surgical resection with safe and active surveillance, focusing on early detection and treatment of regrowth. Although a uniform protocol has not yet been established, a recent consensus recommended an intensified follow-up regimen in the first three years after TNT, consisting in serum CEA evaluation every three months, DRE, rectoscopy, and pelvic MRI every 3–4 months, as well as chest and abdominal CT scans at six months. From the 4th year onwards, a deintensified follow-up regimen may be considered, involving serum CEA evaluation, DRE, rectoscopy, and pelvic MRI every 6 months, along with chest and abdominal CT scans at 12-month intervals [53].

One of the main challenges in the adoption of the watch-and-wait (WW) strategy is the lack of uniform and reproducible criteria for tumor response assessment [54]. In an attempt to standardize the definition of clinical response, Memorial Sloan Kettering developed a three-tiered response/regression schema, which was tested prospectively in the OPRA trial [36,55]. The three tiers are defined based on digital rectal examination (DRE), endoscopy, and MRI (T2W and DWI sequences) and classified as complete, near complete, and incomplete clinical response. A typical cCR is seen as a flat, white scar on endoscopy, without ulceration or nodularity, with only signs of fibrosis on DRE and MRI [55]. The combination of these three modalities could predict the absence of tumors with a reported accuracy of 98% [56]. The value of biopsy samples on residual mucosal abnormalities and additional imaging, such as PET-TC, is still unclear and not recommended routinely [55,57,58].

In order to provide insight into the oncological outcome of WW patients, the International Watch and Wait Database (IWWD) collected and reported the data of more than 1000 patients [59]. In this registry, 25% of patients with a cCR after neoadjuvant therapy and treated with a WW approach developed a local regrowth during the first 2 years of follow-up. Regrowths mostly affected the intestinal wall (97%) and required salvage resection. Nodal local regrowth was very uncommon. Across all patients, 5-year OS was 84.7%, and 5-year disease-specific survival was 93.8%. Patients with a sustained clinical response benefited the most from the WW strategy, with a 5-year OS rate of 87.9%. Patients with local regrowth had a 5-year OS rate of 75.4%. A total of 17.8% of those patients developed distant metastases. Therefore, it has been postulated that the underlying tumor biology, rather than the delayed implementation of immediate surgical intervention, predominantly influences the risk of metastasis and mortality in patients with rectal cancer.

Despite the data heterogeneity, encompassing variations in baseline characteristics, neoadjuvant therapy, and imaging strategies across participating centers, this study provided a reliable representation of the real-world risks and benefits of WW. Rigorous endoscopic surveillance is essential as it facilitates early detection of relapse, allowing for timely curative treatment. The long-term analysis of the OPRA trial demonstrated that 94% and 99% of regrowths occurred within 2 years and 3 years, respectively. These findings highlight the importance of closely monitoring patients during this critical time window [37].

At present, NOM does not constitute a standard of care. However, a growing body of evidence could support its introduction in high-volume centers for selected patients. As healthcare professionals, it is essential to evaluate the potential risks associated with NOM on a patient-by-patient basis, while considering the advantages of organ preservation and its subsequent positive impact on overall quality of life. A thorough and delayed assessment of clinical response could allow sufficient time for tumor biology to manifest itself and therefore better understanding of which patients should be excluded from an NOM strategy.

### 4.3. Role of Circulating Tumor DNA

Since the distant recurrence rate of LARC is still high (estimated at 30%), there is compelling need to find biomarkers that can predict more accurately the risk of recurrence after treatment with radical intent. Moreover, given the variety of alternative therapies, it would be desirable for LARC patients to receive an individualized and biomarker-based treatment. Circulating tumor DNA (ctDNA) has been employed to detect minimal residual disease (MRD) after definitive local treatment in CRC patients [60,61,62]. The GALAXY trial monitored ctDNA status in patients affected by stage II–III and IV resected CRC and demonstrated that ctDNA positivity 4 weeks after surgery was the most significant prognostic factor associated with increased risk of recurrence and worse OS. Conversely, none of the traditionally used clinical and pathological risk factors reached statistical significance [63]. Van Rees et al., in a recent meta-analysis, investigated ctDNA in almost 1600 patients affected by nonmetastatic rectal cancer. After surgery, patients with detectable ctDNA had a significantly increased risk of disease recurrence compared with ctDNA-negative patients (HR 15.54, 95% CI: 8.23–29.34) [64]. Thus far, the likeliness to anticipate a pCR whensoever during multimodality treatment based on ctDNA status has already been recorded [65,66]. 

The current body of research on the effectiveness of ctDNA in LARC should be regarded as limited due to several factors. These factors include the diversity of ctDNA investigations, noticeable in variations such as the collection timeframe, treatment type, and patient follow-up duration, which can potentially misguide the practical application of ctDNA in routine clinical practice. In conclusion, the utilization of ctDNA holds promise in predicting postoperative outcomes and may shed light on the efficacy of additional treatments to improve outcomes following radical surgery in LARC.

However, larger randomized trials are necessary to fully assess the potential role of ctDNA in LARC and ascertain its clinical significance.

## 5. Conclusions

The evidence reported in this review highlights the notable advantages of TNT, including improved pCR, DFS, and OS benefit, tumor downstaging, and potential organ preservation. Assessment of MMR status should be mandated prior to the initiation of neoadjuvant treatment, given its importance in guiding treatment decisions and its potential genetic implications of Lynch Syndrome for both the patient and their relatives.

Here, we present a suggested treatment algorithm for LARC management (Figure 2).

However, it is crucial to consider patient-specific characteristics and engage in comprehensive multidisciplinary team discussions to optimize the selection of appropriate candidates for TNT in collaborative decision-making processes. Moreover, compelling evidence supports the notion that referring patients to high-volume surgeons and centers contributes to improved clinical outcomes. This underscores the significance of seeking specialized expertise and resources in the pursuit of optimal patient care.

Continued research on and clinical investigations into the future incorporation of new biomarkers such as ctDNA, immunotherapy in the context of mismatch repair deficient tumors, and nonoperative management are paving the way for ulterior advancements. These endeavors are essential to further refine and optimize the therapeutic algorithm in LARC management.

## Figures and Tables

**Figure 1 ijms-24-12159-f001:**
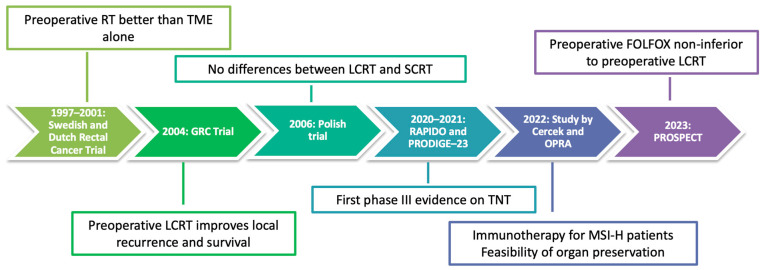
Timeline of treatments in locally advanced rectal cancer.

**Figure 2 ijms-24-12159-f002:**
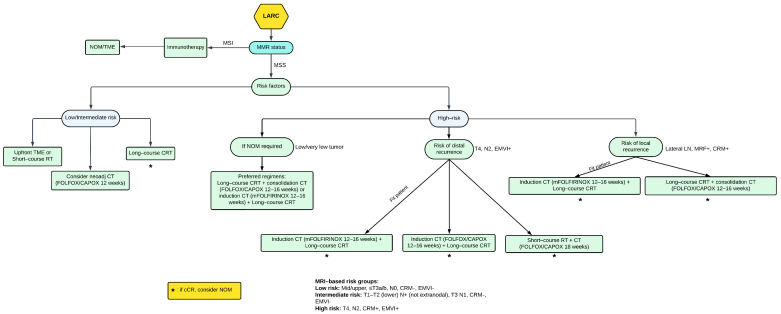
The proposed treatment algorithm for locally advanced rectal cancer: LARC = locally advanced rectal cancer; MMR = mismatch repair; MSI = microsatellite instability; MSS = microsatellite stability; EMVI = extramural vascular invasion; CRM = circumferential resection margin; MRF = mesorectal fascia; LN = lymph nodes; NOM = nonoperative management; TME = total mesorectal excision; LCRT = chemoradiotherapy; CT = chemotherapy; RT = radiotherapy; cCR = clinical complete response.

**Table 1 ijms-24-12159-t001:** Baseline characteristics of patients enrolled in RAPIDO, PRODIGE-23, and PROSPECT trials.

Patients Characteristics	RAPIDO (Exp vs. Std)	PRODIGE-23 (Exp vs. Std)	PROSPECT (Exp vs. Std)
**Number of patients**	462 vs. 450	231 vs. 230	585 vs. 543
**Median age (yrs)**	61 vs. 61	61 vs. 62	57 vs. 57
**Elderly patients (≥65 yrs)**	39% vs. 40%	32% vs. 37%	
**Sex**			
Male	65% vs. 69%	65% vs. 68%	63% vs. 68%
Female	35% vs. 31%	35% vs. 32%	37% vs. 32%
**Performance Status**			
ECOG PS 0	80% vs. 81%	78% vs. 81%	99% vs. 99%
ECOG PS 1	20% vs. 19%	22% vs. 19%	
**Clinical T stage**			
T2	3% vs. 3%	1% vs. 1%	11% vs. 7%
T3	65% vs. 66%	81% vs. 84%	89% vs. 93%
cT4	31.8% vs. 30.4%	18% vs. 16%	/
**Clinical N stage**			
cN2	68% vs. 68%	26% vs. 23%	/
**Distance from anal verge**			
<5 cm	22% vs. 26%	38% vs. 36%	14% vs. 17%
5–10 cm	39% vs. 34%	49% vs. 51%	64% vs. 63%
10–15 cm	32% vs. 34%	13% vs. 13%	22% vs. 20%
Unknown	7% vs. 7%	/	/
**High-risk features**			
EMVI+	32% vs. 28%	Not stated	/
MRF+	60% vs. 62%	26% vs. 27%	/
Lateral N+	14% vs. 15%	Not stated	/

Exp = experimental arm. Std = standard-of-care arm. Yrs = years. cN = clinical nodal. cT = clinical tumor. ECOG = Eastern Cooperative Oncology Group. PS = performance status. N stage = nodal stage. T stage = tumor stage. EMVI = extramural vascular invasion. MRF = mesorectal fascia.

**Table 2 ijms-24-12159-t002:** Clinical outcomes of RAPIDO, PRODIGE-23 and PROSPECT trials after recent updates.

Clinical Outcomes	RAPIDO (Exp vs. Std)	PRODIGE-23 (Exp vs. Std)	PROSPECT (Exp vs. Std)
**Median follow-up**	4.6 yrs	4.6 yrs	4.8 yrs
**Primary endpoint**	3-yrs DrTF	3-yrs DFS	5-yrs DFS
**3-yrs Primary event (Δ%) ***	23.7% vs. 30.4% (6.7%)	76% vs. 69% (7%)	n/a
**5-yrs**	27.8% vs. 34% (7%)	73.1% vs. 65.5% (7.6%)	80.8% vs. 78.6% (2.2%)
**7-yrs**	n/a	67.6% vs. 62.5% (5.1%)	n/a
*** HR (95% CI); *p* value**	0.75 [0.60–0.96]; *p* = 0.019	0.69 [0.49–0.97]; *p* = 0.034	0.92 [0.72–1.14]; *p* = 0.005 for noninferiority
**3-yrs MFS**	80% vs. 73.2%	79% vs. 72%	n/a
**5-yrs**	77% vs. 69.6%	77.6% vs. 67.7%	n/a
**7-yrs**	n/a	73.6% vs. 65.4%	n/a
**pCR rate**	28.4% vs. 14.3%	27.5% vs. 11.7%	21.9% vs. 24.3%
**Local relapse rate**	12% vs. 8% at 5 yrs	5.3% vs. 8.1% at 7 yrs	1.8% vs. 1.6% at 5 yrs
**Distant relapse rate**	23% vs. 30.4% at 5 yrs	20.7% vs. 27.7% at 7 yrs	n/a
**3-yrs OS**	89.1% vs. 88.8%	91% vs. 88%	n/a
**5-yrs OS**	81.7% vs. 80.2%	86.9% vs. 80%	89.5% vs. 90.2%
**7-yrs OS**	n/a	81.9% vs. 76.1%	n/a

Exp = experimental arm. Std = standard-of-care arm. Yrs = years. DrTF = disease-related treatment failure. DFS = disease free survival. HR = hazard ratio. MFS = metastasis-free survival. pCR = pathological complete response. OS = overall survival. n/a = not applicable.

## Data Availability

Not applicable.

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
