# Peer review of "Total Neoadjuvant Treatment for Locally Advanced Rectal Cancer Patients: Where Do We Stand?"

_ijms, 2023, doi:10.3390/ijms241512159_

Round 1

Reviewer 1 Report

Dear Editor,

I found this paper to be a strong, well-executed study.

Please provide more information about the overall strategy for Radiotherapy regimens.

Please provide more information about the pathological outcomes.

Please provide more information about the timing of the surgery.

Please provide more information about the survival outcomes.

 Minor editing of English language required.

Reviewer 2 Report

The authors provide an overview of the current status of total neoadjuvant treatment for locally advanced rectal cancer.

Several issues should still be addressed in the discussion:

- Initially, (as the name suggests) TNT was planned without any adjuvant treatment. Meanwhile, some studies perform adjuvant chemotherapy after TNT + TME. What are the reasons for this? Is there any scientific evidence for this?

- Not all patients with advanced rectal cancer are candidates for TNT. What characteristics are important? Age? Compliance? …

- How common are interruptions or discontinuations during the long duration of TNT treatment?

- How frequently are local progress or distant metastasis observed during the long period of TNT?

Figure A1: Currently, TNT is not the new standard of care. It is an option.

Round 2

Reviewer 1 Report

Dear Editor,

The authors have addressed all my comments for this paper.

Regards

Reviewer 2 Report

The authors have sufficiently addressed the suggestions in the manuscript.